# Effect of Non-Thermal Atmospheric Cold Plasma on Surface Microbial Inactivation and Quality Properties of Fresh Herbs and Spices

**DOI:** 10.3390/foods14213617

**Published:** 2025-10-23

**Authors:** Emel Özdemir, Pervin Başaran, Sehban Kartal, Tamer Akan

**Affiliations:** 1Department of Food Engineering, İstanbul Technical University, 34469 İstanbul, Turkey; basaranakocakp@itu.edu.tr; 2Department of Physics, İstanbul University, 34134 İstanbul, Turkey; sehban@istanbul.edu.tr; 3Department of Physics, Eskişehir Osmangazi University, 26040 Eskişehir, Turkey; akan@ogu.edu.tr

**Keywords:** cold plasma, food quality, food safety

## Abstract

Culinary herbs and spices are highly valued for their contribution to aroma, color, and overall flavor in traditional foods. Microbial inactivation in fresh herbs and spices is challenging due to their complex surface structures and dense natural microflora, which limit the effectiveness of conventional methods. Atmospheric cold plasma (ACP) is an innovative non-thermal technology with potential applications in the fresh spice industry. This study investigates the efficacy of ACP, generated using a practical, simple, and original system that allows uniform treatment without complex equipment, on microbial inactivation and quality attributes of fresh spices. Treatments of 1 and 3 min were applied, and their effects on natural microflora, *Escherichia coli*, and *Pseudomonas syringae* spp. were evaluated on the first day and after 7 days of storage. Results showed that 3 min treatments achieved higher reductions in natural microflora (2.91 log CFU g^−1^), *E. coli* (2.76 log CFU g^−1^), and *P. syringae* spp. (2.24 log CFU g^−1^) compared to 1 min treatments (1.87, 1.93, and 1.65 log CFU g^−1^, respectively). Different herbs exhibited varying responses to ACP, reflecting differences in leaf structure and chemical composition, which highlights the need for tailored treatment strategies. ACP treatment did not significantly affect water activity, color, or moisture content (except for rosemary, bay leaf, and thyme), nor total anthocyanin content (TAA), total phenolic content (TPC), total antioxidant capacity (TAC), or total flavonoid content (TFC). However, total chlorophyll content (TCC) and pH increased significantly in most samples (except rosemary and dill). Scanning electron microscopy (SEM) revealed that the tissue integrity of rosemary and mint was affected by ACP, although more than 50% of carvone in mint was preserved, and its concentration increased. The observed microbial reductions and 3–8-day shelf-life extension suggest meaningful improvements in safety and storage stability for industrial applications. Overall, ACP demonstrates promise as a safe, efficient, and scalable alternative to conventional decontamination methods, with broad potential for enhancing the quality and shelf life of fresh spices.

## 1. Introduction

Fresh herbs and spices are widely used to enhance the flavor, color, and nutritional value of foods and have been valuable commodities since ancient times, with over 2.8 million tons produced annually [1]. To preserve their aromatic and functional properties, these products are often minimally processed or consumed fresh, which increases the risk of microbial contamination [2]. Conventional decontamination methods, including chlorine, ethanol, hydrogen peroxide, ozone, UV light, and organic acids, can extend shelf life but have limitations such as toxic residues, regulatory restrictions, and potential sensory changes [3,4,5,6,7,8]. Atmospheric cold plasma (ACP) has emerged as a promising non-thermal alternative that can inactivate microorganisms without significant temperature increase or toxic residues. Its high scalability makes it particularly suitable for delicate herb structures, where conventional methods may damage tissues or reduce quality [9,10,11]. ACP has been successfully applied to various fresh fruits and vegetables; however, studies on fresh herbs and spices remain limited, creating a knowledge gap regarding their efficacy and safety in these products. Evaluating ACP for herbs such as rosemary, bay leaf, dill, basil, thyme, parsley, mint, and stevia is therefore highly relevant for food safety and shelf-life extension.

The efficacy of ACP depends on factors such as plasma gas composition, electrical parameters, treatment duration, power source, food structure, and relative humidity [12,13,14]. Reactive plasma species, including ions, free radicals, and UV photons, target microbial cell membranes, causing oxidation, electroporation, and cell death [15,16,17]. Using atmospheric air as the plasma gas offers a low-cost, scalable alternative to He, Ar, or N_2_ [18]. This study aimed to address the knowledge gap by evaluating the bactericidal effect of ACP at different exposure times against a spoilage bacterium (*Pseudomonas syringae* ssp.) and a pathogenic bacterium (*Escherichia coli*) on eight fresh herbs and spices. Additionally, this study assessed changes in functional components and the influence of surface properties, such as roughness, on microbial inactivation, using a novel ACP device designed for uniform treatment of delicate herbs.

## 2. Materials and Methods

### 2.1. Atmospheric Cold Plasma System and Its Application

Dielectric Barrier Discharge (DBD) plasma was chosen due to its effectiveness in generating non-thermal reactive species at atmospheric pressure, enabling microbial inactivation and quality preservation in fresh plant products without significant heat damage. DBD plasma was selected based on prior studies demonstrating its efficiency in inactivating microorganisms in fresh herbs and spices and preserving functional compounds [19]. Preliminary tests on mint, rosemary, and thyme also confirmed that 1 and 3 min exposures would avoid excessive heating and tissue damage, justifying the selected treatment durations. DBD plasma provides a uniform plasma distribution over sample surfaces, is scalable for industrial applications, and has been shown to enhance mass transfer, phenolic extraction, and antioxidant retention in plant tissues, as reported in recent studies [19]. A Pacem PTP 22-01 power supply (Pacem Technology A.S., Izmir, Turkey) provided the high-voltage input. The SDBD was produced in the lid of a closed 1 L glass container (Figure 1). A 1.5 mm copper wire was mounted on the inner side of the lid as the high-voltage electrode, while a 2 mm copper plate covering the wire on the outer side of the lid served as the ground electrode. The glass lid was 2 mm thick, and the container had a depth of 18 cm. Atmospheric cold plasma was generated as micro-discharges using a high-voltage supply set at 50 kV and 2500 Hz and applied from 15 cm on samples. Purple-colored plasma formed between the inner copper wire and the lid surface upon energization. The temperature at the top of the glass lid increased noticeably after 3 min of plasma treatment, likely due to the copper electrodes. To prevent overheating during longer treatments, the lid was cooled with a fan after each application. The plasma system was assembled by connecting high-voltage cables to the copper wire beneath the glass and the copper plate above the lid. Fresh spice samples were placed inside the container, and the lid was positioned to fully cover the box. A schematic diagram of the system is shown in Figure 1.

The atmospheric cold plasma air was applied to the rosemary, bay leaf, dill, basil, parsley, mint, thyme, and stevia for different durations of 1 and 3 min. The treatment has been applied separately to each of the samples spread in a thin layer inside the glass storage container. The distance between the plasma source and the sample (15 cm), as well as the power settings (50 kV, 2500 Hz), were based on literature reviews [20,21]. and confirmed by preliminary experiments to achieve microbial reductions without causing tissue damage. In order to evaluate the effectiveness of cold plasma, the system was initially tested on samples for up to 10 min. However, after 5 min, it was observed that the system was warming up samples (e.g., mint, rosemary and thyme) that turned brown and the edges dried. Therefore, the application time was determined as 1 min and 3 min for each sample. Physical and chemical analyses were performed immediately after the atmospheric cold plasma application. Microbial analyses were repeated on the first, third, and seventh day of storage. The results were given by comparing the samples treated with atmospheric cold plasma with the control samples. The samples were stored in airtight polyethylene zip-lock bags with an estimated thickness of 0.02–0.03 mm. These bags exhibit low gas permeability, limiting O_2_ and CO_2_ exchange, thereby helping to maintain moisture and extend shelf life during storage at 4 °C. All experiments were conducted in triplicate to ensure reproducibility. Statistical analyses were performed using SPSS (v.25–28, SPSS Inc., Chicago, IL, USA) and Microsoft Excel (Microsoft Co., Washington, DC, USA). One-way ANOVA was applied to evaluate variance, and results are expressed as mean ± standard deviation. Tukey and Dunnett’s tests were used to identify significant differences between treatments. Pearson correlation coefficients (R^2^) were calculated to determine relationships between microbial inactivation and physicochemical parameters. Statistical significance was considered at *p* < 0.05. These analyses justify the interpretation of microbial and quality changes under different ACP exposure times.

### 2.2. Microbiological Analyses

#### 2.2.1. Preparation of Bacterial Strains, Stock Cultures, and Growth

Bacterial strains were obtained from various culture collections: *Pseudomonas syringae* pv. savastanoi and *Pseudomonas syringae* pv. *phaseolicola* (Istanbul University Culture Collection, *Pseudomonas syringae* spp. (Erzurum Technical University Culture Collection), and *Escherichia coli* 25922 (Istanbul Technical University Culture Collection). Isolates were kept frozen in tryptic soy broth (TSB, Merck Millipore, Darmstadt, Germany) with 30% glycerol at −80 °C. Each isolate of *P. syringae* spp. and *E. coli* 25922 was incubated, respectively, TSB at 30–37 °C, for 24 h. Subsequently, *P. syringae* spp. planted on the tryptic soy agar (TSA) (1.5%) (Merck Millipore, Darmstadt, Germany), and *E. coli* 25922 MacConkey agar and incubated at 30–37 °C, for 48 h.

#### 2.2.2. Sample Inoculation

Rosemary, bay leaf, dill, basil, thyme, parsley, and mint were purchased from MacroCenter in Istanbul, Turkey, and stevia was obtained from Grow Botanik Company, Turkey. *P. syringae* spp. and *E. coli* 25922 were inoculated on fresh spices’ surfaces by dipping technique. Bacterial strains of *P. syringae* (*P. syringae* pv. *savastanoi*, *P. syringae* pv. *phaseolicola*, and *P. syringae* spp.) were cultivated at TSB (Merck Millipore, Darmstadt, Germany) and incubated at 30 °C for 48 h. After incubation, these cultures were all mixed and centrifuged (10 min, 4.000× *g*, 20 °C). *E. coli* 25922 culture mwas cultivated at TSB (Merck Millipore, Darmstadt, Germany) and incubated at 37 °C for 24 h. At the end of the time, in order to measure the turbidity of the cell suspension caused by *E. coli* 25922 and *P. syringae* spp. grown in TSB, the tubes were placed in the McFarland densitometer and adjusted to 10^8^ with sterile peptone physiological saline (8.5 g/L NaCl and 1 g/L peptone). Then, the tubes adjusted to 10^8^ were diluted with peptone physiological saline to prepare a 10^4^ CFU/mL suspension. Afterward, fresh herbs and spices were inoculated by dipping methods into the 50 mL of bacterial suspensions (10^8^ and 10^4^ CFU/mL) for 10 min.

#### 2.2.3. Determination of *P. syringae* spp., and *E. coli* 25922 Counts

*P. syringae spp*. and *E. coli* 25922 inoculated and non-inoculated fresh plant samples (1 g) were placed in centrifuged tubes with 0.1% peptone water 1:9 (*w*:*v*) and mixed by Vortex Shaker (IKA, Staufen, Germany). After mixing, samples were serially diluted with sterile 0.1% peptone water. 100 μL diluted samples were spread-plated on MacConkey for *E. coli* 25922 and TSA for *P. syringae* spp. After incubation in these petri dishes at 37 °C, 24–48 h, respectively, for *E. coli* 25922 and 30 °C, 48 h for *P. syringae* spp., 30–300 colonies were enumerated and expressed as log CFU g^−1^.

#### 2.2.4. Determination of Total Mesophilic Aerophilic Bacteria Count

Total mesophilic aerophilic bacteria (TMAB) count was performed using Plate Count Agar (PCA) by the pour plate method. For this purpose, ~1.0 ± 0.5 g of rosemary, bay leaf, dill, and mint samples were taken and transferred to sterile bags containing 9 mL peptone physiological saline (8.5 g/L NaCl and 1 g/L peptone) and homogenized in a stomacher device (Seward Stomacher^®^, Bohemia, NY, USA) at high speed for 120 s. A total of 1 mL of the diluted samples was transferred to a petri dish and covered with 12–15 mL of PCA medium, then cooled to 45 °C. TMAB count was expressed as log CFU g^−1^ after 48 h of incubation at 37 °C.

### 2.3. Physicochemical Analysis

#### 2.3.1. Weight Loss (%), Color, pH, TAE, Anthocyanin, Flavonoid, Chlorophyll, Total Phenol, Antioxidant, Moisture (%), and Aw

Fresh spices weight loss (WL%) was determined by the difference between the before and after cold plasma spices weights (Wb and Wa, respectively) on the initial day. The result is expressed as a percentage as follows (Equation (1)):Weight loss (%) = ((W_b_ − W_a_)/W_b_ × 100(1)

The color of the spices was measured using a colorimeter (CR-400, Konica Minolta Sensing Inc., Tokyo, Japan) as L*, a*, and b* values. Color readings were taken at three evenly spaced locations on the surface (excluding stems). Following this, ΔE*, BI, H°, and C* values were calculated. Color differences (ΔE*) (between treated and untreated spices) were classified according to scores between 0 and 3 [22]. A value higher than 3 is considered noticeable and obvious to human eyes [23]. According to this classification, the color differences were as follows: (0–0.5) unnoticeable, (0.5–1.5) slightly noticeable, (1.5–3.0) noticeable. ΔE*, hue angle (H°), chroma value, and browning index value (BI) were calculated as Equations (2)–(5) [24,25,26]. The “x” coefficient was used to calculate the BI value [27]. ∆E_ab_* = [(∆L*)^2^ + (∆a*)^2^ + (∆b*)^2^]^1/2^(2)H° = arctan (b*/a*) (3)C* = √(a^2^ + b^2^) (4)BI = ([100(x − 0.31)])/0.17x = [a + (1.75 × L*)]/[(5.645 × L*) + (a* −(3.012 × b*)](5)
pH was measured with a pH meter (Testo 205) at 25 °C, after the samples were homogenized mechanically and diluted with a certain amount of pure water. In the TA analysis, acidity was determined as citric acid and % [28].

Whole fresh spices’ water activity (aw) was measured using a water activity meter (Protimeter, Taunton, UK) at 25 ± 2 °C.

Approximately 1 g of each sample was weighed, and the % moisture content was calculated in a moisture analyzer (Shimadzu MOC63u, Kyoto, Japan).

The physicochemical quality parameters of fresh herbs were analyzed on all atmospheric cold plasma treated and untreated control samples.

#### 2.3.2. Total Phenolic and Antioxidant Content

The total phenolic content (TPC) was determined using the Folin–Ciocalteu method applied by [29]. Gallic acid was used as standard. A total of 100 μL of the extracts diluted in appropriate proportions were transferred to glass tubes and 1.5 mL of 2 N Folin–Ciocalteu (Sigma Aldrich, Schnelldorf, Germany) reagent (diluted 10 times) was added and mixed. Then, 1.2 mL of 7.5% Na_2_CO_3_ (*w*/*v*) solution was added, and the tubes were mixed with a vortex. After the homogenized tubes were kept in a dark environment at 25 °C for 45 min, absorbance values were measured at 765 nm in a UV-Spectrophotometer (UV-1800, Shimadzu). TPC was given as gallic acid equivalent (GAE) using the gallic acid (0–120 mg L^−1^) calibration curve [30]. For total antioxidant content (TAC) analysis, 100 µL trolox and 2 mL 0.1 mM DPPH (1,1-diphenyl-2-picrylhydrazyl) solutions were added to the diluted to a certain extent sample and vortexed. The absorbance values of the samples were kept at room temperature in the dark for 45 min and were measured at 517 nm in a UV-Spectrophotometer. The absorbance of the samples was expressed as % using the Trolox (0–0.025 mg) calibration curve [30].

#### 2.3.3. Total Chlorophyll Content

One g of sample was ground in a mortar with 6 mL of ethanol and then filtered through filter paper (Interlab, Inc., Doral, FL, USA). The filtrates were centrifuged, and the supernatant was transferred to new sterile tubes and made up to 6 mL with extraction solvent. Total chlorophyll content (TCC) was determined by measuring the absorbance values of the extracts diluted in appropriate proportions at 645 nm and 663 nm, respectively. Chlorophyll a, chlorophyll b, and TCC of the samples were calculated as Equations (6)–(8) [31]. Chlorophyll a = 12.7(A663) − 2.69(A645) (6)Chlorophyll b = 22.9(A645) − 4.68(A663) (7)Total chlorophyll content = 20.2(A645) + 8.02(A663)(8)

#### 2.3.4. Total Flavonoid Content

The total flavonoid content (TFC) of substances were determined using catechin as a standard [32]. A total of 0.25 mL of diluted extract was transferred to glass tubes, and 75 μL of 5% NaNO_2_ was added. After the solution was kept for 6 min, 150 μL of 10% AlCl_3_·6H_2_O was added and kept for another 5 min. At the end of the period, 0.5 mL 1 M NaOH and then 2.5 mL pure water were added, mixed by vortex, and absorbance values were measured at 510 nm.

#### 2.3.5. Total Anthocyanin Amount

A colorimetric method, also known as the pH-differential method, was used to determine the total anthocyanin amount (TAA) of the samples. In this method, the total monomeric anthocyanin amount of the samples was determined by taking readings at two different wavelengths (5–700 nm) and two different pH values (1.0–4.5). Firstly, 0.025 M potassium chloride solution (KCl) (pH = 1.0) and 0.4 M sodium acetate (CH_3_COONa·3H_2_O) solution (pH = 4.5) were prepared to be used in the analysis. Pure water was used as the solvent. A 37% HCl acid was used to adjust the solutions to pH 1.0 and pH 4.5. By diluting the extracts with these solutions, a 1 mL dilution was obtained, and the reading was taken after waiting in a dark environment at room temperature for 15 min. The analysis was carried out in three repetitions at two different pH values and two different wavelengths. The absorbance values of the samples were calculated using the formulas below and expressed as mg cyanidin 3-glycoside/100 g. The difference between absorbances was determined as shown in Equation (9).A = (A520 − A700)pH 1.0 − (A520 − A700)pH 4.5(9)

The total amount of anthocyanin was determined as shown in Equation (10).Mg cyn-3-gly equivalents/100 g = (A∗MW∗DF∗100∗10^3∗6)/(ε∗0.25·1000·sample weight)(10)

In Equation (10), molecular weight (MW = 449.2 g/cyanidin for 3-glycoside), dilution factor (DF), length of microplate (0.25 cm), molar extinction coefficient (ε = 26,900 in 1 mol^−1^ ∗ cm^−1^ for cyanidin 3-glycoside), conversion coefficient from g to mg (103), to determine in 100 g of fresh vegetables, initial weight of the sample, amount of solution in which the samples were dissolved (6 mL), and conversion coefficient for converting from ml to L (1000) were used.

### 2.4. Determination of Essential Oil Composition

For this analysis, essential oil samples of fresh mint and rosemary plants were obtained by the 4 h hydrodistillation method in the Clevenger apparatus [33], and the yield (*v*/*w*) was calculated for both peppermint and rosemary oils. After the plant samples for essential oil distillation were cut into small pieces, they were weighed as 100 g, placed in a 2 L flask, and 1000 mL of distilled water was added and heater (MX120, Elektromag, Tekirdağ, Türkiye) was set at 200 °C. The first reading was made 3 h after the water started to boil, the second reading was made 1 h later, and the process was terminated when the 1st and 2nd readings gave the same value. The essential oil accumulated in the Clevenger burette was transferred to the falcon tube and centrifuged. The remaining essential oil was removed with a Pasteur pipette and transferred to amber glass vial tubes and kept in the refrigerator at −20 °C until analysis. Compounds identifications of essential oils were performed by comparison of their mass spectra and retention indices with authentic compounds and literature data [34]. Essential oil components of the obtained peppermint and rosemary oil samples were examined using a Shimadzu GC-MS-QP2020 NX model instrument under the following conditions:

Injection block temperature at 250 °C, detector temperature at 250 °C, carrier gas is He, flow rate is 1 mL/min, temperature of MS source at 280 °C, MS quadrupole temperature at 250 °C, injection mode is split (Split mode 1:10), oven temperature program was at 60 °C for 5 min; at 150 °C for 3 min, increasing by 10 °C per minute from 60 °C to 150 °C, at 300 °C for 5 min, increasing by 5 °C per minute from 150 °C to 300 °C, electron energy 70 eV, mass range 41–400 atomic mass units.

### 2.5. Storage Study

The weight readings of whole leaves of eight untreated herbs and spices weighing approximately 1 g were recorded individually. Other leaves weighing ~1 g from the same package were treated with atmospheric cold plasma for 1 and 3 min. The treated and untreated leaves were then individually sealed in a Ziploc^®^ bag and stored in a refrigerator at 4 °C for 20 days. At the end of the storage period, weight readings were recorded from all leaves. The weight readings were compared with controls (untreated herb and spice leaves) stored under the same conditions.

### 2.6. Scanning Electron Microscope (SEM)

The surface structural changes of fresh mint and rosemary treated with atmospheric cold plasma treatment were analyzed immediately after treatment using SEM according to [35]. The treated/untreated fresh mint and rosemary samples were cut into 1 × 1 cm pieces. The samples were then sputter-coated with gold particles using Emitech K550X Sputter Coating Unit resulting in a coating of 10 nm after 4 min. The samples were examined visually using a JEOL JSM-6390LV (SEMTech Solutions, Inc., North Billerica, MA, USA) at 5 kV. The samples were fixed prior to SEM analysis by immersion in a 2.5% glutaraldehyde solution for 2 h at 4 °C, followed by a graded ethanol dehydration series and air-drying to preserve tissue structure.

### 2.7. Statistical Analysis

All data obtained from repeated measurements were analyzed for variance (one-way ANOVA) using SPSS (v.25–28) Statistical Software (SPSS Inc., USA) and results were expressed as the mean value ± standard deviation. The Pearson correlation cofactor (R2) was used to indicate the correlation. Statistical differences between treated and non-treated fresh spices, atmospheric cold plasma treatment effects, and among exposure times were calculated in terms of microbiological and physicochemical changes by Microsoft Excel (Microsoft Co., USA) and SPSS (v.25–28). The confidence level for statistical significance was 0.05 and it was significant at the *p* < 0.05 level. Tukey and Dunnett’s tests were used to determine the significant difference in data.

## 3. Results and Discussion

### 3.1. Inactivation in Total Mesophilic Aerobic Bacteria (TMAB)

Initial microbial counts revealed that the TMAB load of fresh herbs and spices treated with atmospheric cold plasma for 1 and 3 min was statistically lower than that of the control group (*p* < 0.05) (Figure 2). Maximum inactivation was observed in the rosemary and dill with 2.91 log CFU g^−1^ and 2.54 log CFU g^−1^, respectively, after 3 min of treatment (*p* < 0.05) (Figure 2). 

On the 7th day of storage, 3 min treated stevia and thyme samples showed the lowest counts of 2.44 log CFU g^−1^ and 1.58 log CFU g^−1^, respectively. After 7 days, changes in microbial load on bay leaf, basil, and mint were not statistically significant (*p* > 0.05), indicating herb-specific resistance to ACP treatment. These results demonstrate that TMAB reduction is time-dependent and varies among herbs due to differences in leaf structure and surface properties. The highest reductions in rosemary and dill suggest that denser leaf morphology facilitates the penetration of plasma-generated reactive species, enhancing microbial inactivation. Ref. [36] reported that atmospheric DBD plasma reduced the total microflora load on fresh-cut arugula by 1.02 log CFU g^−1^ after 10 min, which is lower than the reductions observed in the current study. Ref. [37] reported ∼1.0 log CFU g^−1^ reduction on fresh-cut leafy rocket salad after 6 s of surface DBD plasma. The higher reductions in the present study may be attributed to differences in treatment system design, herb matrix, exposure duration, and initial microbial load. Reactive plasma species, including ions, radicals, and UV photons, are responsible for microbial inactivation by damaging cell membranes and inducing oxidative stress. Plant tissue characteristics such as leaf density, roughness, and surface chemistry modulate plasma effectiveness, explaining variability among herb types. TMAB reductions were achieved without significant alterations in water activity, pH, moisture, or bioactive compounds for most herbs, indicating that ACP effectively reduces microbial load while preserving functional quality. ACP treatment for 3 min effectively reduced TMAB in a time- and herb-dependent manner, with rosemary and dill showing the highest reductions. These results align with literature and can be explained by the interactions of plasma-generated reactive species with plant tissues. The treatment preserves physicochemical and bioactive quality, supporting ACP as a safe, non-thermal decontamination strategy for fresh herbs and spices.

### 3.2. Inactivation in Pseudomonas syringae spp.

Microbial counts revealed that the *P. syringae* spp. load of fresh herbs and spices treated with atmospheric cold plasma (ACP) for 1 and 3 min was statistically lower than that of the control group for both 10^4^ and 10^8^ log CFU g^−1^ inoculations (Figure 3 and Figure 4). Increasing the treatment time from 1 to 3 min resulted in significantly higher reductions (*p* < 0.05). For both 10^4^ and 10^8^ inoculations, the highest reductions were observed in rosemary and mint after 3 min of treatment on the first day. Specifically, for 10^4^ inoculations, reductions were 1.59 and 1.75 log CFU g^−1^ on rosemary and mint, whereas for 10^8^ inoculations, reductions were 2.14 and 2.24 log CFU g^−1^ (*p* < 0.05). After 1 min of treatment, minimum reductions were detected on parsley (0.30 log CFU g^−1^ on the first day) and dill (0.24 log CFU g^−1^ on the seventh day). Overall, higher initial microbial loads (10^8^ vs. 10^4^ log CFU g^−1^) led to more pronounced reductions, indicating that inoculum level is critical for ACP effectiveness. These results indicate that microbial inactivation by ACP is both time- and matrix-dependent. Dense leaf structures, such as those of rosemary and mint, likely allow better penetration of plasma-generated reactive species, contributing to higher microbial reductions. Statistically significant reductions (*p* < 0.05) correspond to meaningful decreases in microbial load relevant to food safety. Previous studies report similar or variable reductions depending on plasma type, exposure time, and matrix characteristics. Ref. [2] reported a 5.8 log CFU/g reduction of *Pseudomonas marginalis* on lamb’s lettuce using low-pressure plasma (1.1 kW, 2.45 GHz) in 5 min. Ref. [37] observed 0.29 log CFU/g reduction of *Pseudomonas* spp. on fresh-cut rocket salad using surface DBD plasma for 10 min. Ref. [38] reported that *P. fluorescens* was below detection limit within 3 min plasma jet treatment (20 kV) on lettuce. Ref. [39] also reported a 4.2 log CFU/g reduction in mixed biofilms of *Listeria monocytogenes* and *P. fluorescens* on lettuce leaves after 120 s of atmospheric DBD plasma. Variability among studies is largely due to differences in plasma type, reactive species composition, treatment duration, and matrix properties. Reactive plasma species, including ions, radicals, and UV photons, cause microbial inactivation by disrupting cell membranes and inducing oxidative damage. Plant tissue characteristics, such as surface roughness and wax content, modulate the penetration of these reactive species and explain differences in reduction among herbs. In this study, microbial reductions in *P. syringae* spp. were achieved without major alterations in water activity, pH, moisture, or bioactive compounds in most herbs, suggesting that ACP can reduce pathogens while maintaining product quality. ACP treatment effectively reduced *P. syringae* spp. in a time- and matrix-dependent manner, with rosemary and mint showing the highest reductions. The results align with previous studies and are explained by the interaction of plasma-generated reactive species with microbial cells and plant tissue. Minimal impacts on physicochemical and bioactive properties support ACP as a safe, non-thermal decontamination method for fresh herbs and spices.

### 3.3. Inactivation in Escherichia coli 25922

*E. coli* O157:H7 is one of the most common pathogens in fresh spices, and most of the diseases produced by *E. coli* O157:H7 are associated with biofilms formed on the surface of the food item [40]. Increasing the plasma treatment time from 1 to 3 min resulted in a higher reduction in *E. coli* 25922 load as expected (*p* < 0.05). For both 10^4^ log CFU/mL and 10^8^ log CFU/mL dipping inoculations, the highest reductions were observed in rosemary after 3 min of treatment on the first day (Figure 5 and Figure 6). Specifically, for 10^4^ log CFU/mL inoculation, reductions were 2.07 and 1.72 log CFU g^−1^ on rosemary and mint, respectively, whereas for 10^8^ log CFU/mL inoculation, reductions were 2.76 and 2.28 log CFU g^−1^ on rosemary and stevia. After 1 min of treatment, minimal reductions were detected on parsley (0.59 log CFU g^−1^ on the first day and 0.45 log CFU g^−1^ on the seventh day), highlighting variability among spices. Overall, higher inoculum concentrations (10^8^ vs. 10^4^ log CFU/mL) resulted in more pronounced microbial reductions. These results indicate that microbial reduction is time-dependent and matrix-dependent, with herb structure influencing ACP efficacy. Rosemary, with its dense leaf structure, exhibited the highest reductions, suggesting that plasma-generated reactive species effectively penetrated leaf surfaces and biofilms. The observed reductions were statistically significant (*p* < 0.05) and correspond with a meaningful decrease in microbial load relevant to food safety. Earlier studies reported similar trends: 120 s direct and 2.5 min static DBD plasma application resulted in 2.2 log CFU/g *E. coli* inactivation on lettuce and spinach [39,41]. Ref. [42] observed the highest microbial load reduction (5.5 log CFU/g) on baby kale using a plasma jet for 300 s. Ref. [43] reported a 1.1 log CFU/g decrease in romaine lettuce using in-pack DBD plasma (O_2_-N_2_). Ref. [21] reported 3.77 log CFU/g reduction in baby spinach with 5 min indirect DBD plasma (N_2_). Variability among studies is attributed to differences in plasma type, reactive species composition, voltage, treatment duration, and matrix characteristics. In the current study, differences among herbs likely reflect leaf morphology, surface roughness, and bioactive composition influencing plasma reactivity. Reactive plasma species, including ions, free radicals, and UV photons, are likely responsible for microbial inactivation by damaging cell membranes, inducing oxidation, and triggering electroporation. Plant tissue responses, such as cell wall composition and surface waxes, modulate the penetration and effectiveness of these reactive species, explaining the observed differences among spices. Reductions in *E. coli* corresponded with minimal changes in water activity, pH, moisture, and bioactive compounds in most herbs, indicating that microbial inactivation can be achieved without compromising functional quality. Significant microbial reductions were distinguished from minor functional changes in bioactive compounds, confirming that the treatment is both effective and safe for maintaining quality. ACP treatment for 3 min effectively reduced *E. coli* 25922 loads in a time- and matrix-dependent manner, with rosemary showing the highest reductions. These microbial inactivation results align with prior studies and can be explained by the interaction of reactive plasma species with biofilms and plant tissue structures. The treatment preserved physicochemical and bioactive quality, supporting its potential as a safe, non-thermal decontamination method for fresh herbs and spices.

### 3.4. Assessment of Weight Loss, Aw, pH, TA, and Moisture

Plant intrinsic factors such as weight loss, aw, pH, TA, and moisture are crucial parameters for fresh spice aromatic quality, microbial development, plant deterioration, and final shelf life of the product [44]. Loss of weight due to natural processes of plant tissue transpiration and respiration is the key factor contributing to fresh herbs and spices’ deterioration and their shelf life. Also, weight loss is a vital sign for evaluating the level of fresh spice preservation and dehydration during storage [45]. These physicochemical properties of plasma treated, and non-treated fresh spices were evaluated immediately after treatment, as shown in Table 1. There were no significant differences in weight of fresh spice samples tested in our study with various plasma exposure times (*p* > 0.05). In accordance with our observations, two studies conducted with DBD cold plasma using O_2_-N_2_ and atmospheric air with application times (respectively, 5 min and 10 min) at a distance of 30 mm and 5 mm, respectively, also reported an insignificant change in weight of romaine lettuce [11,43]. In an earlier study, ref. [16] detected a weight loss of 4% and above in radish sprouts treated with low-pressure plasma (N_2_), and it was stated that the change in weight loss may depend on storage temperature and storage duration (4 and 10 °C for 12 days) in that study. Ref. [46] observed no change in the weight of lettuce leaves treated with low-pressure plasma (N_2_) and then at a storage temperature of 4 °C, while a significant weight loss was observed at 10 °C. Although, there was an insignificant change in weight on baby spinach leaves after 2 min of treatment; ref. [47] observed a significant decrease in 14.7 ± 6.3% in weight when compared with untreated control samples after 7-days storage at 4 °C. Storing in vacuumed and plastic zip-lock bags may help prevent moisture and weight loss as this is the case in our study. These diverse observations might be due to plant tissue structure and sensitivity to plasma treatments as well as storage conditions. Potentially, cracks and ruptures formed by plasma treatment on the leaf surface may have caused faster evaporation later with increasing storage durations. In our study, SEM was applied to make observations on the tissue surface after plasma treatment to support this hypothesis.

The aw values of both treatment and non-treatment spices ranged between 89.5 and 97.5, as shown in Table 1. The aw values in treated spices were not significantly different from control samples after 1 and 3 min plasma application (*p* > 0.05). This shows that aw is not affected by atmospheric cold plasma conditions. Studies reported in previous years also did not record any significant change in aw [16,46]. When the moisture content of fresh spices was examined, no significant change was observed in most samples compared to the control, while a slight decrease was detected in rosemary, bay leaf, and thyme (*p* < 0.05) (Table 1).

The pH of a plant product affects sensory and organoleptic features, as well as the composition of microbial flora that might degrade and limit shelf-life [44]. Results of the analysis showed that pH values in treated spices were considerably different and lower than in control samples after 1 min and 3 min atmospheric cold plasma (*p* < 0.05). The pH values of both treated and non-treated spices range between 6.06–6.26, 6.60–6.84, 5.82–6.09, 6.01–6.19, 5.90–6.48, 6.04–6.31, 5.21–5.48, and 5.78–6.06 for rosemary, bay leaf, dill, basil, thyme, parsley, mint, and stevia in order, as shown in Table 1. Reactive nitrogen species such as NO (nitrous oxide), which interact with the moisture in the plant as a result of plasma application, increase the formation of nitric acid, leading to changes in pH and acidity, which can lead to undesirable effects on taste, texture, and shelf life [48]. However, a recent study observed that after applying DBD (100% N) from a distance of 5 cm to baby spinach leaves, the pH decreased from 5.90 to 4.37 and from 4.37 to 2.5 after treatment for 2 and 5 min, respectively, due to the formation of nitric acid (HNO_3_) and nitrous acid (HNO_2_) [21]. In another study conducted on spinach leaves, after 10 min of DBD application, pH increased from 6.42 ± 0.02 to 6.89 ± 0.21 compared to the control samples [47]. Change in pH might be due to the loss of water and metabolic change in the fresh spices after treatment with plasma [49].

Significant changes (from 1.60 to 1.48 (rosemary), from 1.57 to 1.65 (bay leaf), from 1.17 to 1.04 (basil), and from 0.74 to 0.88 (parsley)) were observed when the treatment time increased to 3 min when compared with the control samples (Table 1) (*p* < 0.05). When the application time increased from 1 to 3 min, the TA value in rosemary (from 1.89 to 1.48) and parsley decreased (from 1.08 to 0.88), while in bay leaf and basil, it increased slightly (from 1.43 to 1.65 and from 0.96 to 1.04). As the processing time increases, changes in electrical conductivity may also occur as a result of the increase in organic acid concentration [50]. An increase in H_2_O_2_ may occur because of increased electrical conductivity and acidity [51]. It shows that as the CP application time and energy increases, the O_3_ and nitrogen oxide (NOx) content in the gas phase decreases, and the O_3_ content diffuses into the liquid phase and dissolves in it [52,53].

### 3.5. Color Changes

Bright green color is a critical quality feature for the consumer’s visual acceptance and market value of fresh herbs. The color observations of fresh spices with or without treatment by atmospheric cold plasma at the end of application (1 and 3 min) is shown in Table 1. In our preliminary testing, as the exposure time increased (5, 7, and 10 min), browning was observed in the leaves of fresh spices; therefore, maximum 3 min was selected. For most fresh spices there was no significant difference in L* values between the atmospheric cold plasma-treated and untreated samples (Table 1). Unlike thyme and stevia, other spices did not show a significant increase in ΔE* value after atmospheric cold plasma activation, regardless of the activation time (Table 1). The least color difference (ΔE* = 0.56–0.57) between 1 and 3 min treatment times was found in bay leaves, while the greatest color difference (ΔE* = 0.76–2.92) between 1 and 3 min treatment times was detected in rosemary. Increasing the application time from 1 min to 3 min caused a significant increase in the BI value of mint. Stevia is the least affected spice tested (Table 1). This result confirmed a degree of color change caused by longer plasma exposure; however, it was statistically non-significant (*p* < 0.05). After 3 min of atmospheric cold plasma treatment, a slight increase (*p* < 0.05) was also found in C* (color vibrancy) and H° (color intensity) values in fresh spices. This was consistent with previously reported results for fresh-cut rocket and spinach leaves [36,37,41]. Application distance and time should be taken into consideration for the color quality of fresh spices with high volatile content, such as mint and rosemary. Some studies have reported visible changes in color after the plasma procedure. For instance, an increase in a* value and color differences were reported after treatment in lettuce leaves exposed to corona plasma for 3 and 7 min [35,54]. In another study, it was reported that light areas of spinach leaves treated with DBD plasma became yellowish and dark areas became brownish green after 24 h of storage [55]. This may indicate that the reactive species released as a result of plasma application continue to interact with water and chemical components on the plant surface during storage [56]. Later, ref. [42] detected browning on the edges of cut baby kale (*B. oleracea*) leaves indicating the interaction of polyphenol oxidase and peroxidase enzymes with the substrates.

### 3.6. Assessment of Total Phenol, Antioxidant, Anthocyanin, Chlorophyll, and Flavonoid Content

Spices are preferred not only for improving the flavor of foods; but they go beyond basic dietary requirements and provide health benefits and improve certain body functions due to their high antioxidant and phenolic content [57]. The effects of atmospheric cold plasma treatment on the total phenol, antioxidant, anthocyanin, chlorophyll, and flavonoid content of samples are illustrated in Table 1. The results showed that the total phenolic content (TPC) increased in all fresh spices with the increase in atmospheric cold plasma treatment time. The observed increase in TPC with prolonged cold plasma treatment is likely attributed to structural alterations in plant tissues. Plasma-induced disruptions of cell walls and the epidermis facilitate the release of bound phenolic compounds, leading to higher measurable TPC values. Furthermore, reactive species generated during plasma exposure may enhance the liberation or modification of these compounds, contributing to the overall increase in TPC. In bay leaf, dill, basil, thyme, parsley, and mint, the TPC content of the control sample was 404.60, 219.04, 521.10, 523.85, 574.02, and 380.89 mg GAE/g, and it increased to 416.63, 233.47, 548.25, 598.08, 582.27, and 383.30 mg GAE/g in that order, after 3 min of treatment. In rosemary and dill, the TPC of the control samples, respectively, was 322.82 mg GAE/g and 219.04 mg GAE/g, and it was observed to be reduced to 308.04 mg GAE/g and 213.88 mg GAE/g at the end of the 1 min of treatment. Total antioxidant content (TAC) in mint was observed to increase compared to the control sample at the end of 1 min (5.32–5.41 umol trolox/g). It was observed that the TAC value in the stevia sample increased at the same rate compared to the control sample after 1 (3.56 umol trolox/g) and 3 min treatments (3.57 umol trolox/g), while TAC decreased at the same rate in the dill sample compared to the control sample (4.84–4.79 umol trolox/g). Similarly, total flavonoid content (TFC) of samples other than rosemary, dill, and parsley and mint increased in bay leaf, basil, thyme, and stevia with increasing atmospheric cold plasma treatment time. The greatest decrease in TFC (11.48–10.92 mg EK/g) was seen in the dill after 3 min of exposure. A limited increase was also observed compared to the control sample in rosemary, basil, thyme, and parsley after 3 min (4.16–4.21, 6.68–6.78, 6.88–7.06, and 3.19–3.41 umol trolox/g). When the treatment durations were compared, it was observed that total anthocyanin activity (TAA) in stevia compared to other spices increased more than the control sample after 3 min (16.23–20.84 mg cyanidin 3-glycoside/100 g), while in thyme it decreased after 1 min (21.31–20.37 mg cyanidin 3-glycoside/100 g); however, it showed a slight increase after 3 min (21.78 mg cyanidin 3-glycoside/100 g). In other samples, the amount of TAA increased depending on the increase in the application time. The results were 13.69, 21.84, 18.57, 11.82, 13.83, 18.50 mg cyanidin 3-glycoside/100 g for rosemary, bay leaf, dill, basil, parsley, and mint, respectively. When compared with control samples, these changes in TPC, TAC, TFC, and TAA are not statistically significant (*p* > 0.05). Ref. [58] determined that the flavonoid amount increased after 2 min of low-pressure plasma exposure in freeze-dried lamb’s lettuce while the chlorogenic and caffeic acid concentrations did not change. In addition, they reported that phenolic acids (chlorogenic acid and protocatechuic acid) were degraded more slowly than flavonoids (luteolin and diosmetin) when application time was increased to 5 min [58]. Ref. [4] suggested that reactive species released as a result of plasma exposure can interact with molecules on the leaf surface, causing deformation and rupture of cell bonds, which leads to the release and decomposition of volatile compounds such as water, hydroxide, carbon dioxide, and carbon monoxide out of the cell. Ref. [58] suggested that flavonoids or protocatechuic acid may have increased from flavonoids or other compounds accumulated in epidermal cells as a result of deformation in both the cutin layer and the epidermis as a result of exposure. In another study, it was reported that the antioxidant activity (DPPH) value in lettuce samples applied with low-pressure plasma (10 min) changed around 16.8–20.8% during storage [46]. Whereas, ref. [16] reported that low-pressure plasma (N_2_) did not affect the antioxidant content of fresh-cut radish sprouts. Reactive species released as a result of the application may have caused anthocyanin accumulation in some samples [59]. 

Chlorophyll (Chl) is the main pigment responsible for the color of green leafy spices. ROS released by CP application, as well as plasma related heat, light, pH change, O_2_ derivatives released, and the increased presence of the chlorophyllase enzyme as a result of tissue deformation, may lead to an increased degradation of the chlorophyll pigment [60]. Our results showed that the Chl a and total Chl contents increased with an increase in duration of 3 min for all spices except for rosemary (7.54–6.19 mg/g) and dill (18.36–15.47 mg/g) (*p* < 0.05). The Chl b contents were observed to increase with the increase in plasma application time for other spices except for dill, basil, thyme, and stevia (*p* < 0.05) (Table 1). The maximum reduction in Chl b content was recorded after 3 min of treatment (from 6.05 to 2.71 mg/g) on the stevia sample, whereas the maximum increase in total Chl content was recorded after 3 min treatment (from 20.03 to 28.38 mg/g) on the parsley (*p* < 0.05). The results clearly indicate that atmospheric cold plasma induces the degradation of chlorophyll in some spices. In a recent study, with the pilot-scale generation of plasma processed products, ref. [61] found that chlorophyll degradation resulting in changes in product color and alterations in phenolic compounds were more significant in the continuous mode than in the static plasma treatment mode.

### 3.7. Determination of Essential Oil Components

Common characteristics that define the flavor or quality of the fresh spices and herbs include volatile and other bioactive compounds [62]. The retention time and peak areas of the main volatiles of fresh mint and rosemary samples before and after atmospheric cold plasma treatment are shown in Table 2.

According to our GC/MS analysis, the volatile components that make up peppermint and rosemary essential oils were acyclic monoterpene alcohols (such as linalool, alpha-terpineol, borneol, thymol, alpha-bisabolol, and carvacrol), oxides (such as caryophyllene oxide, limonene oxide, and caryophyllene oxide), monoterpenes (such as limonene, 1,8-cineole, and caryophyllene), alkenes, and aldehydes (such as alpha-pinene, nonanal). The main component in the control and 3 min treated samples of mint was found to be as follows: carvone (51.25% and 62.01%, respectively). While the main components in the control and 3 min treated samples of rosemary were found to be α-pinene (19.64%) and borneol (27.49%), respectively (Table 2). Compared to the control sample, it was observed that atmospheric cold plasma could preserve more than 50% of carvone and increase its amount as a result of a 3 min application. While it was observed that the main component changed as a result of a 3 min application of rosemary. According to our results, in both control samples of mint and rosemary and treated samples, essential oil components were found to be dominated by monoterpenes such as carvone, α-pinene, and borneol, while the treated mint samples were found to be dominated by menthane monoterpenoids such as carvotanacetone. However, major elements and their percentages of samples showed some change after plasma treatment; some aldehydes, terpenes, and alcohols could not be detected because of the 3 min treatment. The losses in some volatile components may have occurred due to damage to the plant tissue due to increased processing time (Figure 7).

Therefore, essential oils are critical for consumer acceptance of both processed foods containing mint and rosemary and for fresh consumption. Our results are partially consistent with the above studies, and plasma application may not be the only reason for the difference in the main components and ratios of the components in the control and atmospheric cold plasma-treated samples. Differences in our study results may be due to the type of cold plasma treatment, its duration, the gas used in plasma production, and the applied voltage, as well as to plant species, surface shape, growing conditions, seasonal conditions, and differences in the active compound of the plant [63,64]. In addition, the data obtained in this study show that each component can be affected at different levels and in different directions depending on the plasma conditions and duration. Early studies reported that there may be losses in essential oil components due to the increase in hydrodistillation time [65]. Ref. [65] reported that the highest volatile compound yield from rosemary in a study comparing supercritical carbon dioxide steam and hydrodistillation was 2.35% in steam distillation and the lowest was 0.35% in water distillation. Although it is not known exactly how cold plasma affects the content and composition of essential oils, the number of studies on its effects on the active ingredients of plants is too low to draw final conclusions. Furthermore, studies conducted in recent years reveal other observations. Ref. [66] reported that a 1 min low-pressure cold plasma treatment caused a 36.7% (0.9 mL/100 g dry matter) increase in lemongrass essential oil content compared to the control, while when the duration was increased from 1 min to 5 min, the essential oil content decreased. In a study conducted on lemon peel, it was reported that the evaporation rate of essential oil components increased due to the increase in plasma treatment time, and as a result, less essential oil was obtained [67]. Ref. [6] reported that the quality of black pepper grains was not affected by 15 and 30 min of non-thermal remote plasma treatment, but there was an insignificant decrease in the amount of the major element piperine. It is also believed that the cold plasma process increases the evaporation rate of essential oil components by increasing surface porosity [63]. The release of volatile compounds from surface modification after plasma application can also contribute to an increase in antimicrobial activity.

### 3.8. Storage Study

When comparing eight control and eight treated samples, the effect of treatment time (1 min and 3 min) on sample weight was found to be unchanged after 20 days of storage in the refrigerator (4 °C). However, on the 12th day of storage, slight yellowing of the leaf edges was observed in the parsley and dill control samples. The results are statistically insignificant, and are not provided. It was observed that cold plasma extended the shelf life of these two samples by 8 days. At the end of the storage period, no weight loss or color change was detected in the stevia and bay leaves treated for 1 and 3 min compared to the control samples. This situation can be attributed to the physiological properties of the plants as well as to the storage in airtight zip-lock bags and appropriate temperature conditions. In addition, no significant weight loss was observed in rosemary, mint, thyme, and basil treated for 1 and 3 min, while slight browning was noted at the tips of leaves in rosemary, thyme, basil, and mint control samples after 13, 12, 7, and 9 days, respectively, and in rosemary, thyme, basil, and mint leaves treated for 1 and 3 min, after 17, 15, 10, and 12 days, respectively (Results not given). This observation is consistent with previous studies reporting minimal weight loss and slight browning in fresh herbs treated under similar conditions [16,46]. However, the decrease in weight values of herbs and spices treated with cold plasma during the storage study was statistically insignificant. This can be explained by the fact that the herb and spice samples were packaged in plastic bags to prevent water loss. Placing the samples in sealed plastic bags with the air removed after plasma treatment may have inactivated enzymes that cause enzymatic browning reactions, such as polyphenol oxidase and peroxidase, which may have led to better preservation and extended shelf life of the processed samples [68,69]. Ref. [42] found that cabbage leaves treated with tap water and stored at 4 °C for 12 days had higher BI values than plasma-treated cabbage leaves, regardless of the treatment time. Ref. [42] also found that plasma-treated cabbage leaves had lower BI, chroma, and b* values compared to control samples, while there was no change in a* values. Ref. [16] stored radish leaves at 4 °C and 10 °C for 12 days after treating them with N_2_-cold plasma for 0, 2, 5, 10, and 20 min. They reported that the moisture content decreased as the treatment time increased, but its appearance and odor did not change. In another study by [37], in which they evaluated the quality and shelf life of arugula salad treated 10 min with atmospheric cold plasma, they reported that the samples of shelf life stored at 2, 5, and 9 °C increased by 53, 27, and 18 h, respectively, compared to unprocessed ones. Studies have shown that cold plasma treatment under optimum conditions and appropriate storage temperature has the potential to increase shelf life and product quality during refrigerated storage.

### 3.9. Electron Microscopy

Compositional changes and differences among all spices after atmospheric cold plasma indicated that during plasma treatment surface shape and/or leaf structure may have been affected. Deformation in the surface structure due to the increase in application time may have affected microbial inactivation and the number of volatile components [15]. The structure of rosemary and mint samples before and after atmospheric cold plasma processing was studied using scanning electron microscopy (SEM) images, taken at 5000× magnification (Figure 7). After atmospheric cold plasma treatment, samples showed similar structure, i.e., voids and the presence of cracks compared to control samples, as shown in Figure 7B–D’. Compared to the control, the most severe damage to rosemary and mint samples was found in the area after 5 min of treatment (Figure 7D,D’). Ref. [35] reported that lettuce leaves showed similar rough surfaces and stomata as compared to the control samples after corona plasma application. The formation of micro-cracks on leaf surfaces during cold plasma treatment results from physical and chemical changes induced by the plasma. Reactive oxygen and nitrogen species (ROS and RNS), high-energy electrons, and UV radiation interact with plant tissues, causing oxidative degradation of cell wall components and the epidermis. These interactions lead to micro-level surface cracks, which can increase leaf surface hydrophilicity, potentially enhancing water absorption and biological activity. However, prolonged plasma exposure may compromise the mechanical integrity of plant tissues and affect quality. Therefore, careful optimization of plasma treatment time and parameters is essential [61]. Ref. [70] reported cracks, ruptures, cell ablation, roughness, and granularity on the surface of green tea leaves treated with cold plasma (N_2_). To compare our results to [70]’s results, the increase in the duration of atmospheric cold plasma and the interaction of reactive substances such as RONS, which are released by the chemical decom-position of free radicals, with the surfaces of plants and spices prove that they can cause damage to the surface microstructure of plant cells. It has been reported that voids, surface roughness, and cracks may form as a result of the interaction between reactive oxygen species (ROS) generated during cold plasma treatment and the rupture of hydrogen and non-covalent bonds within the cell wall structure [71]. In the study by [71], untreated grape pomace exposed to dielectric barrier discharge (DBD) plasma for 5 minutes showed a significantly thinner and less distinct cell wall structure. When the treatment duration was extended to 15 minutes, the cell wall was observed to disappear completely. Ref. [72] studied the effect of DBD cold plasma assisted in microwave treatment of betel leaves and suggested that it caused cracks, fissures, and cellular damage. Ref. [73] reported that many pores and cracks formed on the surface of wood apple peels treated with DBD plasma for 20 min, and this would increase the extraction efficiency as it would increase the surface hydrophilicity of the material. Recent studies have reported that reactive species released as a result of cold plasma application can damage the cell wall in plant cells through oxidation, electroporation, and the mechanical force of the shock wave [74,75,76,77].

## 4. Limitations and Future Perspectives

While atmospheric cold plasma shows great potential for enhancing the quality and microbial safety of fresh spices, its large-scale industrial application remains limited due to high economic costs, unless more cost-effective and sustainable systems are developed. The findings highlight the potential of ACP for industrial-scale applications and commercialization, providing a non-thermal alternative to conventional sterilization methods. Since research on fresh spices is still relatively scarce, further investigations are required to clearly understand and optimize the effects of specific processing parameters in relation to different product types. Particular attention should be given to studying interactions between plasma-generated reactive species and bioactive compounds to avoid degradation of functional components. Before plasma-treated fresh spices can be introduced for consumption, it is essential to clarify how the free radicals generated during treatment interact with the biological components of the products. Optimization of ACP parameters for different spice types is crucial to balance microbial inactivation and quality retention. Reactive species released during plasma production may induce oxidative and enzymatic changes, potentially resulting in browning or loss of freshness; therefore, treatment duration, power, and frequency must be carefully controlled. Future studies should also evaluate long-term storage stability and shelf-life effects under real-world conditions. If these processes can be optimized, cold plasma could become a preferable alternative to traditional sterilization methods. Cost-effective approaches and comparisons with conventional sterilization techniques should be explored to facilitate industrial adoption. Additionally, standardization and reproducibility across different plasma systems and equipment scales, as well as safety and regulatory considerations, must be addressed before large-scale commercial implementation.

## 5. Conclusions

Atmospheric cold plasma (ACP), a non-thermal technology, effectively inactivates TMAB, *E. coli* 25922, and *P. syringae* spp. on fresh spice surfaces. Treatments of 3 min were significantly more effective than 1 min exposures for microbial decontamination and stability during storage (*p* < 0.05). The extent of microbial inactivation was influenced by plasma power, treatment duration, and microorganism resistance. ACP also increased the shelf life of fresh herbs and spices by 3–8 days during 20 days of storage at 4 °C, without markedly affecting most physicochemical properties, except for slight increases in pH, titratable acidity, moisture content, and chlorophyll levels. Surface properties, such as roughness, may influence microbial inactivation rates and warrant further investigation. Higher plasma power, longer treatment duration, and increased frequency can degrade oxidation-sensitive pigments and bioactive compounds. Therefore, plasma parameters must be carefully optimized to minimize quality loss in fresh spices. The findings highlight the potential of ACP for industrial-scale applications and commercialization, offering a non-thermal alternative for enhancing food safety and shelf life of fresh herbs and spices. However, this study has limitations, including the need to evaluate a broader variety of spice types, different initial microbial loads, and real-world processing conditions. Future research should focus on optimizing plasma parameters to balance microbial inactivation with quality retention and on standardizing plasma systems to ensure reproducibility across different equipment scales. Safety and regulatory considerations must be addressed before large-scale implementation of ACP in commercial settings.

## Figures and Tables

**Figure 1 foods-14-03617-f001:**
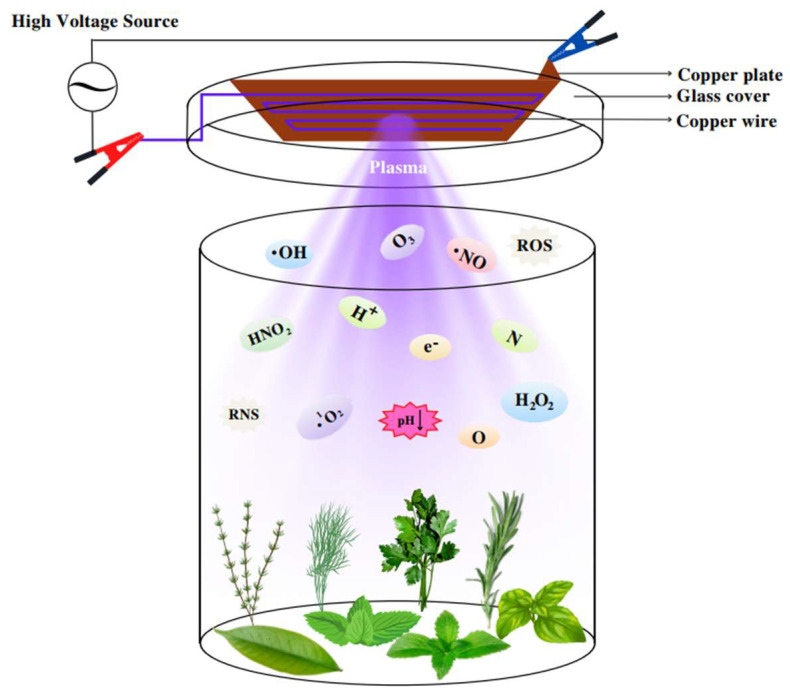
Schematic diagram of the atmospheric cold plasma system.

**Figure 2 foods-14-03617-f002:**
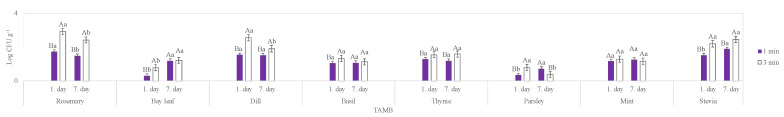
Inactivation of TAMB in fresh herbs and spices after 1 and 3 min of treatment using atmospheric cold plasma at selected voltage (50 kV, 2500 Hz) and air flow. Upper- and lower-case letters indicate significantly different exposure times (on the same day) and storage days (on each exposure time), respectively (Tukey test, *p* < 0.05). Error bars represent the standard deviation.

**Figure 3 foods-14-03617-f003:**
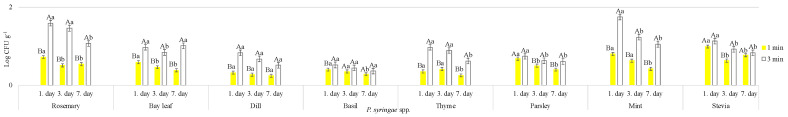
Inactivation of *P. syringae* spp. cocktail in fresh herbs and spices after 1 and 3 min of treatment using atmospheric cold plasma at selected voltage (50 kV, 2500 Hz) and air flow (initial cell count, 10^4^ CFU g^−1^). Upper- and lower-case letters indicate significantly different exposure times (on the same day) and storage days (on each exposure time), respectively (Tukey test, *p* < 0.05). Error bars represent the standard deviation.

**Figure 4 foods-14-03617-f004:**
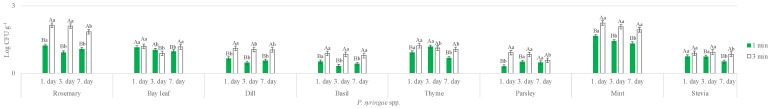
Inactivation of *P. syringae* spp. cocktail in fresh herbs and spices after 1 and 3 min of treatment using atmospheric cold plasma at selected voltage (50 kV, 2500 Hz) and air flow (initial cell count, 10^8^ CFU g^−1^). Upper- and lower-case letters indicate significantly different exposure times (on the same day) and storage days (on each exposure time), respectively (Tukey test, *p* < 0.05). Error bars represent the standard deviation.

**Figure 5 foods-14-03617-f005:**
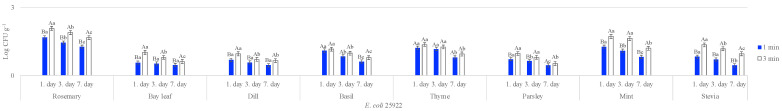
Inactivation of *E. coli* 25922 in fresh herbs and spices after 1 and 3 min of treatment using atmospheric cold plasma at selected voltage (50 kV, 2500 Hz) and air flow (initial cell count, 10^4^ CFU g^−1^). Upper- and lower-case letters indicate significantly different exposure times (on the same day) and storage days (on each exposure time), respectively (Tukey test, *p* < 0.05). Error bars represent the standard deviation.

**Figure 6 foods-14-03617-f006:**
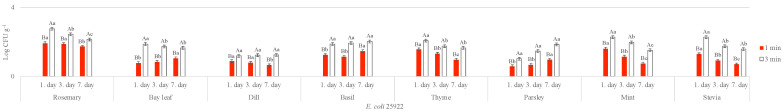
Inactivation of *E. coli* 25922 in fresh herbs and spices after 1 and 3 min of treatment using atmospheric cold plasma at selected voltage (50 kV, 2500 Hz) and air flow (initial cell count, 10^8^ CFU g^−1^). Upper- and lower-case letters indicate significantly different exposure times (on the same day) and storage days (on each exposure time), respectively (Tukey test, *p* < 0.05). Error bars represent the standard deviation.

**Figure 7 foods-14-03617-f007:**
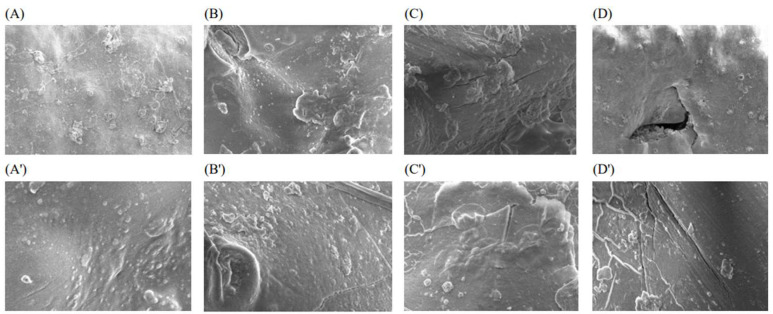
Surface micrographs of rosemary and mint samples at magnification level 5000× level, (**A**,**A’**) (0 control), (**B**,**B’**) (1 min), (**C**,**C’**) (3 min), and (**D**,**D’**) (5 min), respectively.

**Table 1 foods-14-03617-t001:** Effect of atmospheric cold plasma treatment on the physicochemical properties of fresh spices on the initial day.

Analysis	Plasma Treatment Time (min)	Samples							
		Rosemary	Bay Leaf	Dill	Basil	Thyme	Parsley	Mint	Stevia
Moisture %	0 (Control)	79.68 ± 0.09 ^a^	44.69 ± 2.11 ^b^	85.61 ± 0.37 ^a^	78.37 ± 0.13 ^a^	77.05 ± 0.58 ^a^	84.59 ± 0.32 ^a^	84.98 ± 0.22 ^a^	79.77 ± 0.63 ^a^
	1	74.45 ± 0.45 ^b^	46.97 ± 2.69 ^a^	86.02 ± 0.16 ^a^	77.68 ± 0.21 ^a^	73.81 ± 0.18 ^b^	85.62 ± 0.35 ^a^	84.39 ± 0.68 ^a^	80.01 ± 0.18 ^a^
	3	75.62 ± 0.31 ^b^	45.39 ± 1.78 ^b^	85.47 ± 0.69 ^a^	78.21 ± 0.40 ^a^	75.19 ± 0.28 ^b^	86.05 ± 1.17 ^a^	83.84 ± 0.08 ^a^	79.52 ± 0.48 ^a^
aw	0 (Control)	90.5 ± 0.71 ^a^	94.5 ± 2.12 ^a^	94.0 ± 1.41 ^a^	97.0 ± 1.41 ^a^	92.0 ± 1.41 ^a^	95.0 ± 0.00 ^a^	97.5 ± 0.71 ^a^	92.0 ± 1.41 ^a^
	1	91.5 ± 0.71 ^a^	95.5 ± 0.71 ^a^	94.0 ± 0.00 ^a^	96.0 ± 1.41 ^a^	91.5 ± 0.71 ^a^	94.0 ± 1.41 ^a^	96.5 ± 0.71 ^a^	92.5 ± 0.71 ^a^
	3	89.5 ± 0.71 ^a^	96.5 ± 0.71 ^a^	92.5 ± 0.71 ^a^	96.5 ± 0.71 ^a^	92.5 ± 0.71 ^a^	95.0 ± 1.41 ^a^	97.0 ± 0.00 ^a^	93.5 ± 0.71 ^a^
pH	0 (Control)	6.26 ± 0.03 ^a^	6.84 ± 0.03 ^a^	6.09 ± 0.02 ^a^	6.19 ± 0.02 ^a^	6.48 ± 0.09 ^a^	6.31 ± 0.03 ^a^	5.48 ± 0.08 ^a^	6.06 ± 0.01 ^a^
	1	6.20 ± 0.07 ^ab^	6.72 ± 0.03 ^b^	5.99 ± 0.01 ^b^	6.13 ± 0.01 ^a^	6.24 ± 0.11 ^b^	6.13 ± 0.05 ^b^	5.47 ± 0.03 ^b^	5.98 ± 0.01 ^b^
	3	6.06 ± 0.08 ^b^	6.60 ± 0.01 ^c^	5.82 ± 0.07 ^c^	6.01 ± 0.07 ^b^	5.90 ± 0.03 ^c^	6.04 ± 0.03 ^c^	5.21 ± 0.02 ^b^	5.78 ± 0.02 ^c^
Total Acidity	0 (Control)	1.60 ± 0.08 ^ab^	1.57 ± 0.04 ^ab^	1.06 ± 0.05 ^a^	1.17 ± 0.0 ^a^	2.13 ± 0.16 ^a^	0.74 ± 0.09 ^b^	0.99 ± 0.05 ^a^	1.35 ± 0.05 ^a^
	1	1.89 ± 0.04 ^a^	1.43 ± 0.63 ^b^	0.92 ± 0.02 ^a^	0.96 ± 0.04 ^b^	2.06 ± 0.05 ^a^	1.08 ± 0.02 ^a^	0.91 ± 0.11 ^a^	1.16 ± 0.09 ^a^
	3	1.48 ± 0.13 ^b^	1.65 ± 0.21 ^a^	0.91 ± 0.11 ^a^	1.04 ± 0.03 ^ab^	2.16 ± 0.03 ^a^	0.88 ± 0.03 ^ab^	1.12 ± 0.09 ^a^	1.39 ± 0.07 ^a^
TPC (mg GAE/g)	0 (Control)	322.82 ± 15.02 ^a^	404.60 ± 18.59 ^a^	219.04 ± 16.11 ^a^	521.10 ± 10.93 ^a^	523.85 ± 13.03 ^a^	574.02 ± 21.45 ^a^	380.89 ± 17.75 ^a^	644.12 ± 24.15 ^a^
	1	308.04 ± 6.27 ^a^	413.88 ± 15.00 ^a^	213.88 ± 9.30 ^a^	531.07 ± 5.19 ^a^	542.06 ± 10.72 ^a^	579.18 ± 11.34 ^a^	386.73 ± 18.15 ^a^	679.18 ± 14.89 ^a^
	3	314.23 ± 8.99 ^a^	416.63 ± 18.17 ^a^	233.47 ± 10.38 ^a^	548.25 ± 32.59 ^a^	598.08 ± 16.67 ^a^	582.27 ± 11.89 ^a^	383.30 ± 16.32 ^a^	639.31 ± 13.05 ^a^
TAC (umol trolox/g)	0 (Control)	4.16 ± 0.06 ^a^	7.19 ± 0.26 ^a^	4.84 ± 0.06 ^a^	6.68 ± 0.02 ^a^	6.88 ± 0.09 ^a^	3.19 ± 0.31 ^a^	5.32 ± 0.03 ^a^	3.56 ± 0.02 ^a^
	1	4.15 ± 0.03 ^a^	7.15 ± 0.07 ^a^	4.79 ± 0.04 ^a^	6.73 ± 0.07 ^a^	7.01 ± 0.04 ^a^	3.30 ± 0.19 ^a^	5.41 ± 0.14 ^a^	3.57 ± 0.04 ^a^
	3	4.21 ± 0.03 ^a^	7.12 ± 0.05 ^a^	4.79 ± 0.05 ^a^	6.78 ± 0.02 ^a^	7.06 ± 0.11 ^a^	3.41 ± 0.05 ^a^	5.32 ± 0.01 ^a^	3.57 ± 0.03 ^a^
Chl a	0 (Control)	7.54 ± 0.26 ^a^	8.81 ± 0.30 ^b^	18.36 ± 0.14 ^a^	11.63 ± 0.35 ^b^	14.18 ± 0.34 ^a^	14.67 ± 0.91 ^c^	11.96 ± 0.11 ^c^	15.30 ± 0.27 ^b^
	1	7.52 ± 0.23 ^a^	8.79 ± 0.30 ^b^	16.24 ± 0.26 ^b^	11.72 ± 0.70 ^b^	14.62 ± 0.27 ^a^	17.61 ± 0.31 ^b^	16.02 ± 0.47 ^a^	13.55 ± 0.38 ^c^
	3	6.19 ± 0.34 ^b^	10.77 ± 0.91 ^a^	15.47 ± 0.26 ^c^	14.11 ± 0.56 ^a^	15.47 ± 0.99 ^b^	20.71 ± 0.19 ^a^	13.23 ± 0.42 ^b^	21.20 ± 0.74 ^a^
Chl b	0 (Control)	3.15 ± 0.15 ^b^	2.27 ± 0.38 ^b^	6.72 ± 0.81 a	6.01 ± 1.27 ^a^	6.20 ± 1.08 ^a^	5.36 ± 0.26 ^c^	4.58 ± 0.05 ^c^	6.05 ± 0.13 ^b^
	1	3.19 ± 0.04 ^b^	3.21 ± 0.11 ^a^	5.84 ± 0.07 ^b^	5.78 ± 0.87 ^a^	5.43 ± 0.11 ^a^	6.14 ± 0.11 ^b^	5.78 ± 0.17 ^a^	5.48 ± 0.17 ^b^
	3	3.68 ± 0.54 ^a^	3.83 ± 0.25 ^a^	5.62 ± 0.07 ^c^	5.08 ± 0.21 ^a^	5.79 ± 0.37 ^a^	7.68 ± 0.09 ^a^	5.09 ± 0.10 ^b^	2.71 ± 0.39 ^a^
Total Chl (a + b)	0 (Control)	10.69 ± 0.34 ^ab^	11.07 ± 0.08 ^b^	25.08 ± 0.22 ^a^	17.64 ± 0.92 ^a^	20.39 ± 1.17 ^b^	20.03 ± 1.18 ^c^	16.55 ± 0.16 ^c^	21.35 ± 0.40 ^b^
	1	10.71 ± 0.27 ^a^	12.10 ± 0.41 ^b^	22.08 ± 0.33 ^b^	17.50 ± 0.42 ^a^	20.05 ± 0.38 ^b^	23.75 ± 0.42 ^b^	21.80 ± 0.63 ^a^	19.03 ± 0.54 ^c^
	3	9.87 ± 0.39 ^b^	14.59 ± 1.15 ^a^	21.09 ± 0.33 ^c^	19.18 ± 0.77 ^b^	21.26 ± 1.36 ^a^	28.38 ± 0.26 ^a^	18.32 ± 0.51 ^b^	23.91 ± 1.13 ^a^
TAA (mg cyanidin 3-glycoside/100 g)	0 (Control)	13.23 ± 1.71 ^a^	19.91 ± 0.90 ^a^	16.50 ± 1.22 ^a^	10.55 ± 1.80 ^a^	21.31 ± 0.31 ^a^	10.89 ± 1.29 ^a^	16.37 ± 1.48 ^a^	16.23 ± 1.71 ^a^
	1	13.56 ± 1.10 ^a^	20.11 ± 0.41 ^a^	17.10 ± 1.10 ^a^	10.55 ± 2.06 ^a^	20.37 ± 0.76 ^a^	12.69 ± 2.90 ^a^	16.97 ± 2.58 ^a^	18.77 ± 0.95 ^a^
	3	13.69 ± 0.58 ^a^	21.84 ± 1.75 ^a^	18.57 ± 2.52 ^a^	11.82 ± 0.69 ^a^	21.78 ± 1.50 ^a^	13.83 ± 0.53 ^a^	18.50 ± 1.67 ^a^	20.84 ± 1.06 ^a^
TFC (mg EK/g)	0 (Control)	9.91 ± 0.29 ^a^	27.87 ± 0.38 ^a^	11.48 ± 0.09 ^a^	2.49 ± 0.03 ^a^	11.17 ± 0.40 ^a^	9.47 ± 0.27 ^a^	2.56 ± 0.072 ^a^	9.50 ± 0.04 ^a^
	1	9.86 ± 0.04 ^a^	27.96 ± 1.08 ^a^	11.41 ± 0.58 ^a^	2.50 ± 0.04 ^a^	11.17 ± 0.34 ^a^	10.10 ± 0.61 ^a^	2.45 ± 0.04 ^a^	9.29 ± 0.06 ^a^
	3	9.78 ± 0.02 ^a^	28.22 ± 0.88 ^a^	10.92 ± 1.08 ^a^	2.51 ± 0.06 ^a^	11.25 ± 0.45 ^a^	9.97 ± 0.32 ^a^	2.46 ± 0.02 ^a^	9.88 ± 0.06 ^a^
L*	0 (Control)	45.94 ± 3.19 ^a^	47.09 ± 1.27 ^a^	38.54 ± 0.60 ^a^	43.02 ± 0.77 ^a^	38.64 ± 1.92 ^a^	44.03 ± 1.18 ^a^	35.92 ± 0.62 ^a^	38.44 ± 2.40 ^a^
	1	46.41 ± 2.49 ^a^	46.63 ± 1.10 ^a^	38.74 ± 1.58 ^a^	42.54 ± 1.50 ^a^	40.45 ± 2.03 ^a^	43.77 ± 1.32 ^a^	36.94 ± 0.71 ^a^	38.08 ± 1.96 ^a^
	3	48.81 ± 2.14 ^a^	46.29 ± 0.88 ^a^	39.10 ± 0.81 ^a^	420.0 ± 1.73 ^a^	39.81 ± 1.50 ^a^	43.58 ± 0.65 ^a^	36.64 ± 1.09 ^a^	38.64 ± 2.36 ^a^
ΔE*	0 (Control)	-	-	-	-	-	-	-	-
	1	0.76	0.71	0.56	0.54	2.11	0.39	1.06	0.71
	3	2.92	0.99	0.57	1.31	1.81	0.76	2.07	0.57
BI	0 (Control)	-	-	-	-	-	-	-	-
	1	32.60	24.68	59.30	71.88	40.13	41.15	33.13	18.05
	3	32.16	22.02	54.51	72.17	44.56	42.76	42.77	18.22
H°	0 (Control)	55.86	48.10	55.40	55.53	56.76	55.14	51.28	51.76
	1	55.22	49.10	55.27	55.89	59.10	55.69	51.30	49.87
	3	56.41	47.33	55.34	55.28	59.96	55.55	53.86	50.69
C*	0 (Control)	26.27	29.47	31.19	38.06	24.15	29.16	24.97	18.32
	1	26.79	29.29	31.41	38.07	23.67	29.24	25.25	18.34
	3	26.72	29.89	31.09	38.89	24.37	29.73	26.52	17.91

Mean ± S.D. Values followed by the same letter in the same column are non-significant at *p* < 0.05 according to Tukey’s HSD test.

**Table 2 foods-14-03617-t002:** Volatile components identified in mint and rosemary before and after atmospheric cold plasma treatment by GC–MS.

	Sample	Plasma Treatment Time						
	Mint	0 (Control)			3 min			
ID	Volatile components	RI	RT	AREA %	RI	RT	AREA %	LRI
1	2-Hexenal, (E)-	843	4.4	0.21	-	-	-	817–844
2	3-Hexen-1-ol, (Z)-	-	-	-	845	4.5	0.60	829–862
3	α-Pinene	935	6.7	2.20	935	6.7	0.61	924–951
4	Sabinene	973	7.7	2.41	973	7.7	1.16	958–981
5	β-Pinene	978	7.8	3.96	978	7.8	1.34	962–987
6	β-Myrcene	988	8.1	1.78	988	8.1	1.18	975–991
7	3-Octanol	996	8.3	3.81	996	8.3	2.52	981–1005
8	α-Terpinene	1017	8.7	0.14	1017	8.7	0.35	1001–1024
9	Limonene	1031	9.1	11.23	1031	9.0	6.34	1012–1038
10	1,8-Cineole	1035	9.1	5.25	1035	9.1	4.41	1013–1039
11	β-Ocimene, (Z)-	-	-	-	1047	9.4	0.33	1028–1047
12	1,3,8-p-Menthatriene	1059	9.7	0.28	1059	9.7	0.85	1074–1118
13	Sabinene hydrate, trans-	1072	9.9	0.34	1072	9.9	2.26	1070–1107
14	Terpinolene	-	-	-	1086	10.2	0.19	1064–1091
15	Linalool acetate	1098	10.5	0.62	1098	10.5	0.29	1234–1254
16	p-Mentha-2,8-dien-1-ol, trans-	1123	10.9	0.09	-	-	-	1122–1142
17	Limonene oxide, trans-	1139	11.2	0.13	1139	11.2	0.10	1126–1149
18	Menthol	1173	11.8	1.77	1174	11.8	1.62	1172–1182
19	Limonen-4-ol	1183	12.0	0.89	1184	12.0	3.27	1167–1189
20	Dihydrocarvone, trans-	1200	12.3	4.16	1199	12.3	1.48	1162–1206
21	Verbenone	1211	12.5	0.19	-	-	-	1190–1224
22	Carveol, cis-	1229	12.8	0.64	1222	12.7	0.43	1196–1224
23	Carvone	1257	13.2	51.25	1258	13.2	62.01	1227–1265
24	Menthyl acetate	1322	14.3	0.20	1322	14.3	0.25	1278–1310
25	Elemene	1334	14.4	0.15	1334	14.4	0.16	1418–1499
26	Piperitenone	1340	14.6	0.22	-	-	-	1329–1347
27	Carvyl acetate, cis-	-	-	-	1355	14.8	0.56	1310–1366
28	β-Bourbonene	1390	15.4	1.56	1390	15.4	1.58	1366–1400
29	Caryophyllene, (Z)-	1426	16.1	2.17	1425	16.1	0.70	1392–1426
30	β-Cubebene	1433	16.3	0.25	1433	16.3	0.37	1370–1560
31	γ-Cadinene	-	-	-	1448	16.6	0.59	1490–1521
32	γ-Muurolene	1448	16.6	0.74	-	-	-	1455–1494
33	Muurola-4 (14),5-diene, cis-	-	-	-	1466	17.0	1.02	1448–1478
34	α-Humulene	1460	16.9	0.84	-	-	-	1430–1466
35	Germacrene-D	1486	17.5	0.83	1487	17.5	3.53	1458–1491
36	Bicyclogermacrene	-	-	-	1502	17.8	0.37	1474–1501
37	γ-Elemene	1502	17.8	0.34	-	-	-	1418–1499
38	Calamenene, cis-	1524	18.3	0.68	1525	18.3	0.63	1492–1528
39	Spathulenol	1582	19.7	0.17	-	-	-	1562–1590
40	Caryophyllene oxide	1588	19.9	0.36	-	-	-	1563–1595
41	Cubenol	1619	20.6	0.14	1619	20.6	0.16	1600–1644
42	α-Muurolol	-	-	-	1658	21.6	0.21	1620–1656
	Rosemary	0 (Control)			3 min			
ID	Volatile components	RI	RT	AREA %	RI	RT	AREA %	LRI
1	α-Thujene	927	6.5	0.63	927	6.5	0.33	916–938
2	α-Pinene	935	6.7	19.64	935	6.7	8.34	924–951
3	Camphene	951	7.1	5.33	952	7.1	2.65	936–965
4	β-Pinene	978	7.8	8.56	978	7.8	4.66	962–987
5	β-Myrcene	988	8.1	2.07	988	8.1	1.35	975–991
6	α-Phellandrene	1005	8.5	0.81	1005	8.5	0.61	990–1009
7	3-Carene	1008	8.6	0.94	1008	8.6	1.54	1001–1010
8	α-Terpinene	1017	8.7	1.43	1017	8.7	0.53	1001–1024
9	m-Cymene	1025	8.9	2.90	1025	9.0	1.41	998–1037
10	Limonene	1030	9.0	3.24	1030	9.0	2.35	1012–1038
11	1,8 Cineole (Eucalyptol)	1034	9.1	11.28	1035	9.1	7.07	1013–1039
12	Sabinene hydrate, trans-	1072	9.9	0.38	1072	9.9	0.24	1070–1107
13	α-Terpineol	1086	10.2	1.11	1086	10.2	1.20	1178–1203
14	Linalool	1098	10.5	2.67	1099	10.5	5.01	1074–1098
15	Nonanal	-	-	-	1103	10.6	0.13	1093–1118
16	p-Menth-2-en-1-ol, trans-	-	-	-	1127	11.0	0.26	1095–1130
17	Verbenol, cis-	1145	11.3	0.21	1146	11.4	0.51	1110–1146
18	Camphor	1151	11.4	6.91	1152	11.4	7.28	1106–1153
19	γ-Terpinene	-	-	-	1169	11.8	0.46	1049–1069
20	Borneol	1179	11.9	1.80	1179	12.0	27.49	1152–1177
21	Pinocamphone, cis	1180	12.0	1.82	1181	12.0	2.51	1162–1180
22	Terpinen-4-ol	1183	12.0	2.83	-	-	-	1165–1189
23	α-Terpinyl acetate	1197	12.3	5.39	1198	12.3	5.22	1324–1348
24	Verbenone	1211	12.5	4.98	1213	12.5	5.16	1190–1224
25	Carvotanacetone	1247	13.1	0.32	-	-	-	1230–1256
26	Bornyl acetate	1287	13.7	11.76	-	-	-	1259–1284
27	α-Cubebene	-	-	-	1348	14.7	0.18	1334–1379
28	Jasmone	-	-	-	1362	14.9	0.28	1359–1379
29	α-Copaene	-	-	-	1380	15.2	0.45	1360–1392
30	Geranyl acetate	1371	15.1	0.92	-	-	-	1355–1370
31	Methyl eugenol	1397	15.5	1.81	1398	15.5	1.68	1364–1402
32	Caryophyllene	1425	16.1	2.94	1426	16.1	3.64	1392–1426
33	Geranyl acetone	-	-	-	1442	16.5	0.78	1422–1453
34	α-Humulene	1460	16.9	1.23	1460	16.9	2.24	1430–1466
35	α-muurolene	-	-	-	1502	17.8	0.46	1477–1502
36	γ-Cadinene	-	-	-	1518	18.2	0.72	1490–1521
37	δ-Cadinene	1521	18.3	0.45	1522	18.3	1.31	1498–1526
38	Caryophyllene oxide	1588	19.9	1.22	1589	19.9	1.33	1563–1595
39	α-Bisabolol	-	-	-	1686	22.2	0.64	1649–1686

(RI: Retention Indices, RT: Retention Time, LRI: Literature Retention Indices).

## Data Availability

The original contributions presented in this study are included in the article. Further inquiries can be directed at the corresponding author.

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
