# Peer review of "Effect of Non-Thermal Atmospheric Cold Plasma on Surface Microbial Inactivation and Quality Properties of Fresh Herbs and Spices"

_foods, 2025, doi:10.3390/foods14213617_

Round 1
Reviewer 1 Report
Comments and Suggestions for Authors
General Comment: The manuscript refers to atmospheric cold plasma on surface microbial inactivation and quality properties of eight fresh herbs and spices. This subject concerns a scientifically relevant topic. The article is well-written, with appropriate methodology and recent citations. Numerous quality analyses were performed, and the data were well discussed. However, the review was hampered by the inability to locate the manuscript's figures and tables.
Line 111 and Line 127: The authors did not present Figure 1 anywhere in the manuscript.
Line 201: The authors refer to the term "irradiation." In my opinion, this is not the most appropriate term to refer to the exposure of spices to cold plasma.
Line 361: The authors did not present the manuscript's figures and tables.
Line 749: The authors should include the limitations of cold plasma technology in the discussion section and point out perspectives for future studies.
Line 750: The conclusion should be more succinct.
Reviewer 2 Report
Comments and Suggestions for Authors
Overall Assessment
This article addresses the current and important topic of cold plasma technology in extending the shelf life of fresh herbs and spices. The work contains extensive research material, but requires language corrections, shortening some method descriptions, and clarifying the interpretation of results.
________________________________________
Lines Needing Improvement
1. Abstract
Line 16: "Culinary herbs and spices are highly desirable for taste development, mostly adhering to the flavoring..." – the sentence is ungrammatical. It should be amended to, for example: "Culinary herbs and spices are highly desirable for taste development, mostly contributing to the flavoring and coloring of traditional foods."
Lines 28–29: "pH values in treated spices were considerable different..." – should be "considerably different."
Lines 30–31: repetition: "Total flavonoid content... increased... An induced degradation of chlorophyll was observed..." – requires a more coherent transition.
2. Introduction
Line 52–54: "conventionally" → correct to "conventionally."
Line 64: "Irradition" → correct to "Irradiation."
Line 68–70: "application, which can inactivate target microorganisms without causing a significant temperature increase... and it does not cause toxic residue formation" – the sentence is too long, I suggest splitting it into two.
Line 72: "Atmospheric cold plasma have been successfully used..." – should read "has been successfully used."
Materials and Methods
Line 108–117: The description of the plasma system is chaotic, with repetitions (e.g., about the thickness of glass and copper). It requires simplification and a clearer drawing.
Line 172: missing units – it should be indicated that these are units of bacterial concentration (CFU/mL).
Line 204: The equation (Eq. 1) has poor formatting – the subscripts and fractional signs need to be corrected.
Lines 106-137
Problem: repetitions ("glass lid designed to produce atmospheric cold plasma was placed to completely cover the box...").
Recommendation: shorten the description, leaving key information (parameters: 50 kV, 2500 Hz, 1–3 min). Lines 224–225 (formulas)
o Problem: Formulas written inconsistently (partly with formatting errors, e.g., "[(5.645𝑥𝐿)+(𝑎 −(3.012x𝑏]")).
o Recommendation: Correct mathematical notation according to journal standards.
Results – Microbiology (lines 357–371)
o Problem: Lack of clear summary after describing numerical values; some sentences are grammatically incorrect ("…was successful in reducing the total viable count a similar at a level of approximately…").
o Recommendation: Correct grammar and add one summary sentence.
Line 365–366: "(P>0.05) Dimitrakellis et al. (2021)..." – missing period before the quote.
Lines 470–478 (pH)
o Problem: Expression "considerably different" – linguistic error.
o Recommendation: Change to "significantly different."
Lines 498–528 (color changes)
o Problem: Repeating "Table 1" and excessive numerical comparisons.
o Recommendation: Summarize, leaving the most important differences (e.g., that rosemary and mint are most susceptible to color changes).
Lines 595–681 (essential oils)
o Problem: Very long sentences and repetition of chemical names.
o Recommendation: Move detailed ingredient lists to supplementary materials; leave only the most important compounds in the main text (e.g., carvone, α-pinene, borneol).
5. Lines 682–699 (storage study)
o Problem: Frequent repetition of "statistically insignificant" and "Results not provided."
o Recommendation: Instead, provide a short summary and limit the details.
________________________________________
Key general recommendations
• Shorten and simplify the abstract and Introduction.
• Correct linguistic errors ("considerable different," "have been used" → "has been used").
• Standardize formulas and table formatting.
• Add stronger summaries in the results sections to provide the reader with clear, practical conclusions.
The language in this article is generally understandable, but it has several serious shortcomings:
Grammatical and stylistic errors – incorrect forms are common, e.g., "considerable different" instead of "considerably different," or ungrammatical sentences ("mostly adheres to the flavoring..." in the abstract).
Repetition and verbosity – the phrase "cold plasma treatment" appears frequently, making the text difficult to read. Many paragraphs repeat the same information instead of summarizing the results concisely.
Terminological and unit inconsistencies – the units used vary (e.g., log CFU g-1 vs. log CFU/g), and chemical symbols and Greek letters are sometimes written in words (a-pinene instead of α-pinene).
Formatting – equations and tabular data are interwoven throughout the text, hindering readability. Organized tables would be preferable.
Style – In places, it resembles a literal translation from Turkish/English, with calques and unstable constructions.
In summary, the article requires a thorough linguistic overhaul – shortening, simplifying, and standardizing the style to make the text clearer and bring it up to the standard of an international publication.
Reviewer 3 Report
Comments and Suggestions for Authors
Thank you for your submission. I noticed that the manuscript was not prepared in the required format. Please revise and submit your paper according to the journal’s submission guidelines.
Additionally, in the Results section, several figures, tables, and other elements are mentioned but not included in the manuscript. Kindly ensure that all referenced figures, tables, and supporting materials are included in your revised submission.
Once the manuscript is corrected to include all figures and tables and formatted properly, I will go ahead and provide my full feedback.
Thank you for your attention to these points.
Reviewer 4 Report
Comments and Suggestions for Authors
The subject matter is noteworthy, and the manuscript is well-organized and well-written. However, I would like to see a few issues addressed to improve its quality:
-
The abstract is too lengthy and should be rewritten in a more concise and compact form.
-
The references should be updated by replacing older sources (before 2015) with more recent studies.
-
Line 109: The reasons for choosing DBD should be explained. You may refer to the following publication: https://doi.org/10.1007/s11947-023-03096-z
-
Line 141: The specifications of the bags used should be reported, including thickness and permeability to CO₂ and O₂.
-
Section 2.6: How was the structure of the samples fixed prior to SEM analysis?
-
Line 537: Please explain why the TPC increased with longer treatment duration.
-
Lines 693–694: This statement should be supported with a reference.
-
Lines 726–727: The reasons for the formation of micro-cracks on the surface of the leaf should be discussed. You may refer to the following publication: https://doi.org/10.1080/07373937.2022.2050255
-
The conclusion is too lengthy and should be rewritten in a more compact and concise manner, focusing only on the most important findings. In addition, I recommend that the authors provide suggestions for future studies.
Round 2
Reviewer 1 Report
Comments and Suggestions for Authors
Comment: In line 151 the authors kept the term "Irradiation"
Comment: Figure 1 should be redone. This Figure is not informative. What was the distance from the cold plasma emission point to the spices? What was the mass of herbs used? How were these herbs arranged in the treatment system? Was it just the leaves?
Comment: I recommend that authors use a real photo of the treatment system.
Comment: The quality of Figures 2; 3; 4; 5 and 6 needs to be improved.
Table 1- I recommend that the authors remove the data for a* and b*. These data presented in isolation are not relevant.
Author Response
"Please see the attachment."

Reviewer 2 Report
Comments and Suggestions for Authors
Good morning,
Recommendation for the future.
If you make corrections, please provide the lines containing the corrected sections in your response to the reviewer's comments. We provide the line numbers in the corrected manuscript. Providing lines from the old version of the manuscript makes it difficult to check the corrections.
Author Response
"Please see the attachment."

Reviewer 3 Report
Comments and Suggestions for Authors
Consider revising the title to focus on the ACP system, its non-thermal nature, and its role in improving microbial safety and quality.
Abstract
Clearly explain why microbial inactivation in fresh herbs and spices is challenging and why ACP is a promising alternative to conventional methods.
Emphasize what is practical, simple, and original about the ACP system and how it improves on existing technology.
Summarize how different herbs responded to ACP treatment and discuss what these differences mean rather than listing scattered exceptions.
Indicate whether microbial reductions and the 3–8 day shelf-life extension are sufficient from a safety or industrial perspective.
Strengthen the conclusion by highlighting the broader implications for the spice industry and potential applications.
Ensure all bacterial species, including E. coli and P. syringae, are italicized.
Introduction
Highlight the knowledge gap and explain why ACP studies on fresh herbs and spices are limited and important for food safety.
Emphasize the novelty and significance of the ACP system early, especially its suitability for delicate herb structures.
Clearly state the research objectives, including microbial inactivation, functional quality, and surface property evaluation.
Improve logical flow by organizing content from context → limitations of conventional methods → ACP potential → knowledge gap → study objectives.
Materials and Methods
Clearly report all experimental parameters, including units, voltages, frequencies, distances, temperatures, and durations.
Provide justification for selected ACP and microbial parameters, referencing prior studies or preliminary experiments.
Specify statistical tests, number of replicates, significance thresholds, and how correlations were interpreted.
Results and Discussion
Clearly describe trends over time and quantify microbial, physicochemical, and bioactive changes.
Link microbial reductions and quality changes to plasma-generated reactive species and plant tissue responses.
Report sample sizes, p-values, and units consistently.
Compare results critically with the literature, explaining similarities or differences based on matrix, plasma conditions, or microbial type.
Organize subsections logically: observation → significance → literature comparison → mechanism → summary.
Integrate microbial inactivation results with physicochemical and bioactive data for a cohesive discussion.
Distinguish statistically significant changes from functionally meaningful bioactive changes.
Add summary statements at the end of each subsection emphasizing key findings.
Ensure all bacterial species are italicized throughout.
Please include pictures of samples before and after treatment to visually demonstrate effects.
Limitations and Future Perspectives
Emphasize the need for studies on interactions between plasma-generated reactive species and bioactive compounds.
Discuss optimization of ACP parameters for different spice types.
Suggest cost-effective approaches for large-scale applications and comparisons with conventional sterilization methods.
Include assessments of long-term storage stability and shelf-life effects.
Conclusions
Highlight broader practical implications of ACP for industrial-scale applications and commercialization.
Discuss study limitations, including the need for diverse spice types, variable microbial loads, and real-world processing conditions.
Recommend future research on optimizing plasma parameters to balance microbial inactivation and quality retention.
Address standardization and reproducibility across different plasma systems and equipment scales.
Highlight safety and regulatory considerations before large-scale implementation.
Ensure all bacterial species names are consistently italicized throughout the manuscript.
Author Response
"Please see the attachment."
